# Effect of Membrane Orientation and Concentration of Draw Solution on the Behavior of Commercial Osmotic Membrane in a Novel Dynamic Forward Osmosis Tests

**DOI:** 10.3390/membranes12040385

**Published:** 2022-03-31

**Authors:** Du Bai, Boguslaw Kruczek

**Affiliations:** Department of Chemical and Biological Engineering, University of Ottawa, Ottawa, ON K1N 6N5, Canada; dbai047@uottawa.ca

**Keywords:** forward osmosis, dynamic performance test, time lag, thin film composite membrane, membrane orientation

## Abstract

Dynamic performance tests, commonly used to characterize gas separation membranes, are not utilized to characterize osmotic membranes. This paper demonstrates the application of a novel dynamic forward osmosis test to characterize a commercial osmotic membrane. In particular, we report the effect of membrane orientation (active layer draw solution (AL-DS) vs. active layer feed solution (AL-FS)) and the draw solution concentration on the membrane’s transient and steady-state behaviors. A step-change in the draw solution concentration initiated the dynamic test, and the mass and concentration of the feed and draw solutions were recorded in real-time. The progress of the experiments in two different membrane orientations is markedly different; also, the draw solution concertation has a different effect in the orientations. A positive salt time lag is observed in both orientations; however, the salt time lag in the AL-FS orientation (4.3–4.6 min) is practically independent of the draw solution concentration, but it increases from 7 to 20 min with the draw solution concertation in the AL-DS orientation. A negative water time lag, ranging from −11 to −20 min depending on the draw solution concentration, is observed in the AL-DS orientation. Still, in the AL-FS orientation, the water flux is practically constant from the experiment’s onset, leading to a negligible water time lag (<1 min). The new method demonstrated in this paper can be a potent tool for characterizing osmotic membranes.

## 1. Introduction

Forward osmosis (FO) is a membrane process that has gained immense interest, particularly in the last decade, with potential applications in desalination, wastewater treatment and reclamation, bioproducts and the food industry, and energy generation and resource recovery [1,2,3,4,5]. FO utilizes the osmotic pressure difference of solutions across a semipermeable membrane to draw water from a dilute feed solution (FS) to a more concentrated draw solution (DS); the salt moves in the opposite direction. Compared to pressure-driven membrane processes such as reverse osmosis (RO), nanofiltration (NF), and ultrafiltration (UF), FO processes are less energy-intensive, show less membrane fouling, and FO membranes are easier to clean [1,4,6].

The major challenge associated with FO processes is inadequate water fluxes due to inherent internal concentration polarization (ICP) within the porous membrane layer. As a result, much research is focused on developing improved FO membranes [7]. The performance of FO membranes is typically characterized in terms of the following three intrinsic parameters: water permeability (*A*), solute permeability (*B*) and structural parameter (*S*) [8]. The *A* and *B* describe the transport across the selective layer of the membrane, and *S* quantifies the resistance to the mass transport across the membrane support layer, which is responsible for the ICP [9]. Substituting these three parameters into governing transport equations predicts the water flux (*J_w_*) and reverse salt flux (*J_s_*) of a membrane in FO experiments.

The most common approach to evaluating *A*, *B,* and *S* utilizes the RO-FO method [10]. First, *A* and *B* are measured in a reverse osmosis test, i.e., by applying hydraulic pressure to an aqueous solution of a solute, typically an inorganic salt. Then, knowing *A* and *B*, *S* is determined in a subsequent FO experiment, with the salt used in the RO test as a draw solute. The FO test can be carried out in the following two different membrane orientations: the active layer facing feed solution (AL-FS) or the active layer facing draw solution (AL-DS). In principle, the same values of *S* should be obtained regardless of the membrane orientation [9].

In practice, FO membranes are characterized by different feed and draw solutions at different operating conditions and different testing apparatus. It makes an objective comparison of different FO membranes challenging. Consequently, Cath et al. [10] proposed standardizing the experiments used to determine the intrinsic properties of FO membranes. They specified the operating conditions (including the solute type and its concentration, the temperature, the pressure, the crossflow velocity, etc.) at which the tests should be performed. Independent characterization of the same membranes at the same standardized conditions by eight independent labs led to comparable *A*, *B,* and *S* [10]. However, the standardized protocol, or generally combined RO-FO approach, requires multiple experiments in different experimental setups. More importantly, this approach assumes that *A* and *B* are independent of the nature of the driving force, the direction of water and solute transport, and the negligible effect of the applied hydraulic pressure on the properties of the active and support layers of the membrane. However, *B* determined from RO and FO tests is different [11].

Tiraferri et al. [12] proposed an alternative methodology for characterizing FO membranes that does not require RO tests to determine *A* and *B*. They determined *A*, *B,* and *S* from the governing transport equations for *J_w_* and *J_s_* using nonlinear least-squares regression analysis. They obtained the required multiple sets of *J_w_* and *J_s_* values in the AL-FS orientation by gradually increasing the draw solution concentration, i.e., using a multi-stage protocol [13]. The method was also adapted to FO tests with AL-DS orientation or, more generally, pressure retarded osmosis (PRO) tests with different pressures of the draw solution [14]. Bilad et al. [13] also obtained different *J_w_* and *J_s_* values using multiple filtration and single-stage protocols. The former required a new membrane for each concentration of the draw solution. The latter involved a single batch FO test during which the concentration of the draw solution decreased significantly, allowing the measurement of *J_w_* and *J_s_* at different concentrations of the draw solution. The determined *A*, *B,* and *S* depend on the governing transport equations for *J_w_* and *J_s_* [15]. Tiraferri et al. [12] assumed the applicability of the Van’t Hoff equation and negligible dilutive external concentration polarization (ECP) in the draw solution. Still, they accounted for the concentrative ECP in the feed solution. Martin et al. [16] also considered all mass transfer boundary layers and the non-ideality of concentrated draw solutions (i.e., the Van’t Hoff equation was not required). On the other hand, Kim et al. [17], in addition to the applicability of the Van’t Hoff equation, also assumed a negligible concentrative ICP in the AL-DS orientation. It allowed calculation of *A* and *B* from explicit equations using experimentally measured *J_w_* and *J_s_* in the FO test with AL-DS orientation.

The RO-FO and FO methods rely on steady-state experiments. In contrast, gas permeation membranes commonly rely on dynamic experiments to determine the membrane’s time lag [18]. The latter is a measure of the resistance to gas transport across the membrane [19]. The ICP and ECP, which significantly affect the performance of FO membranes, are also the measures of the respective mass transfer resistances. However, dynamic experiments are not practiced when characterizing osmotic membranes. The primary barrier is initiating a dynamic experiment, which requires a step-change in driving force. We have recently demonstrated the possibility of dynamic FO experiments initiated by a step-change in the draw solution concentration. We then monitored changes in the mass of draw and feed solutions and the salt concentration in the feed solution [20]. The AL-DS water transport showed a negative time lag, similar to dynamic gas permeation experiments based on pressure decay monitoring. On the other hand, salt transport showed a positive time lag analogous to dynamic gas permeation experiments based on pressure rise monitoring [21]. 

This work aims to systematically study the effects of membrane orientation (AL-DS vs. AL-FS) and the magnitude of change in the draw solution concentration on the dynamic behaviour of a commercial osmotic membrane. In addition to the water flux and the reverse salt flux, these effects are quantified using experimentally observed time lags associated with the water and reverse salt transport. The transport of water and salt in FO membranes is far more complex than the gas transport in polymer films, for which the time-lag method is applicable. The relevant equations for the quantitative analysis of the dynamic test of FO membranes have yet to be developed. Therefore, herein we attempt to provide a physical interpretation of the observed results in the framework of the existing models for RO and FO membranes. To our best knowledge, this is the first time the effects of membrane orientation and concertation of the draw solution are reported in the literature using the concept of dynamic FO experiments. 

## 2. Theoretical Background

The models describing water and salt transport in osmotic membranes fall into two categories based on the assumed structure of the active layer—microporous or nonoporous [22]. The first category includes finely porous (FP), surface force-pore flow (SF-PF), modified surface force-pore flow (MD-SF-PF), and preferential sorption-capillary flow (PS-CF) models. The second category includes the solution-diffusion (SD), the extended solution-diffusion (ESD), and the solution-diffusion-imperfection (SDI) models [22]. The SD is the most commonly used transport model [23].

### 2.1. Reverse Osmosis

According to the SD model, the rate equation for the water flux ($J_{w})$ in RO membranes is given by the following: (1)$$J_{w}=A\left(\Delta p-\Delta \pi \right)$$
where *A* is the water permeability coefficient and Δ*p* and Δ*π* are the hydraulic and osmotic pressure gradients across the membrane, respectively. The osmotic pressure for a dilute solution of an electrolyte is given by the following:(2)π=icRT
where: *i* is the number of ionic species each molecule will dissociate, *c* is the molar concertation of solute, *R* is the universal gas constant, and *T* is the absolute temperature. 

The rate equation for the salt flux (*J_s_*), on the other hand, is given by the following: (3)$$J_{s}=B\left(c_{m}-c_{p}\right)$$
where: *B* is the solute permeability coefficient, respectively, and *c_m_* and *c_p_* are the solute concentrations at the membrane surface on the feed side and permeate side, respectively. Typically, *c_m_* is greater than *c_f_*; the latter being the solute concentration in the bulk feed solution. The situation in which *c_m_* > *c_f_* indicates the occurrence of an external concentration polarization (ECP), which is undesirable. 

The SDI model extends the SD model by allowing the solvent and solute transport through the defects (i.e., leakage of the solution through the membrane). Consequently, the rate equations for the water and solute transport are modified to the following: (4)$$J_{w}=A\left(\Delta p-\Delta \pi \right)+K_{3}\Delta p$$
(5)$$J_{s}=B\left(c_{m}-c_{p}\right)+K_{3}\Delta pc_{m}$$
where: *K*_3_ is the coupling coefficient, which signifies the flow through the defects.

### 2.2. Forward Osmosis

The basis for transport and separation in FO processes is the difference between the osmotic pressure across a semipermeable membrane resulting from the difference in solute concentration in the draw solution (*c_d_*) and the feed solution (*c_f_*). In FO processes, the solvent (water) is transported from the feed solution into the draw solution across the membrane, implying that *c_d_* > *c_f_*. In the absence of hydraulic pressure gradient, assuming the applicability of the SD model, the rate equation for the water flux is given by the following: (6)$$J_{w}=A\left(\pi_{d,m}-\pi_{f,m}\right)$$
where, subscript *m* signifies that the osmotic pressures of the draw and feed solutions are based on the solute concentrations, *c_d,m_* and *c_f,m_*, on the respective membrane surfaces. The osmotic pressures *π_d,m_* and *π_f,m_* could be significantly different from the respective bulk osmotic pressures *π_d,b_* and *π_f,b_*. 

Unlike RO processes, the solute flux (*J_s_*) in FO processes moves opposite to the water flux. Assuming applicability of the SD model, the solute flux, which is also referred to as the reverse solute flux, is given by an equation similar to Equation (3) as follows:(7)$$J_{s}=B\left(c_{d,m}-c_{f,m}\right)$$

The opposite directions of the water and solute fluxes give rise to an inherent internal concentration polarization (ICP). Unlike the ECP, which can be eliminated by creating turbulence near the membrane surface to facilitate the transport of the rejected solute from the membrane surface back to the bulk solution, the ICP cannot be eliminated because it occurs within the pores of the support layer of the membrane. On the other hand, since there is no applied hydraulic pressure, the ECP in FO processes is much less severe than in pressure-driven RO processes [24]. 

Forward osmosis systems can be operated in the following two different membrane orientations: the active layer facing draw Solution (AL-DS) and the active layer facing feed solution (AL-FS). In the AL-DS orientation, *π_f,m_* > *π_f,b_* and the resulting concentration polarization is referred to as a concentrative ICP. On the other hand, in the AL-FS orientation, *π_d,m_* < *π_d,b_* and the resulting concentration polarization is referred to as dilutive ICP. The extent of the ICP, in either case, depends on the solute resistivity (*R′*) in the porous support [9]:(8)$$R^{\prime }=\frac{L_{s}}{D_{eff}}$$
where *L_s_* is the thickness of the porous support and *D_eff_* is the diffusivity of solute in the porous substructure. If *L_s_* is very small and/or *D_eff_* is very large, *π_f,m_* → *π_f,b_* in the AL-DS orientation and *π_d,m_* → *π_d,b_* in the AL-FS orientation, which maximizes the water flux. The water flux is inversely related to the solute resistivity, which is mathematically expressed by [9]:(9)$$J_{W}=\frac{1}{R^{\prime }}ln\left(\frac{\pi_{d,b}}{\pi_{f,b}}\right)$$

The diffusivity of solute in porous substructure is always smaller than solute diffusivity in bulk solution (*D*) as follows: (10)$$D_{eff}=\frac{D\varepsilon_{eff}\delta }{\tau }$$
where *τ* and *ε_eff_* are the tortuosity and effective porosity of the support layer, respectively, and *δ* is the constrictivity factor, which is related to the ratio of the molecular solute diameter (*d_s_*) to the pore diameter (*d_p_*) of the support:(11)$$\delta ={\left(1-\frac{d_{s}}{d_{p}}\right)}^{4}$$

Substituting Equation (10) into Equation (8) leads to the following alternative expression for the solute resistivity:(12)$$R^{\prime }=\frac{L_{s}\tau }{D\varepsilon_{eff}\delta }=\frac{S}{D\delta }$$
where *S* is a structural parameter that characterizes the porous substructure of an asymmetric membrane as follows:(13)$$S=\frac{L_{s}\tau }{\varepsilon_{eff}}$$

It is important to note that sometimes the constrictivity factor *δ* does not appear in Equation (12) [25], which implies it is close to unity. The latter is the case when *d_s_* << *d_p_*.

### 2.3. Time-Lag Method

A dynamic gas permeation experiment performed in a constant volume (CV) system provides the basis for the time lag method [19]. The experiment is initiated by a step increase in pressure at the feed side of the membrane (initially, both sides of the membrane are at high vacuum). The resulting pressure rise at the permeate side and/or pressure decay at the feed side of the membrane are monitored (Figure 1). 

The time lag (*θ*) is the intercept of the linear portion of the pressure response curves with the time axis (Figure 1). The time lag tests are typically performed using single gases; however, they can also be carried out using gas mixtures [27]. 

The time-lag is inversely proportional to the diffusion coefficient of the gas in the membrane as follows:(14)$$\theta_{u}=-\frac{L^{2}}{3D}$$
(15)$$\theta_{d}=\frac{L^{2}}{6D}$$
where *L* represents the membrane thickness and subscripts *u* and *d* denote the membranes’ upstream (feed side) and downstream (permeate side). The permeability coefficient (*P*) is directly proportional to the slope of the linear portion of the pressure response curve in Figure 1. According to the following SD model:(16)P=KD

Therefore, knowing *P* and *D*, the solubility coefficient (*K*) can be evaluated by rearranging Equation (15). In other words, the time-lag method estimates the three basic transport properties in the SD model from a single dynamic gas permeation test.

## 3. Materials and Methods

Low-pressure RO TFC membranes from Toray Industries Inc. were used in all tests. According to the manufacturer (Toray), they had the water flux rejection of a 500 mg/L aqueous NaCl feed solution at 110 psig, 25 °C of 32.7 L·m^−3^·h^−1^ and 99%. Although RO TFC membranes are not ideal in FO applications, this membrane was selected because of the expected large ICP arising from the presence of nonwoven support.

The schematic of the testing system is presented in Figure 2 [20]. The membrane was installed in a Teflon crossflow symmetric membrane cell (CF016D-FO Cell, Sterlitech, Auburn, WA, USA) with an effective permeation area of 20.6 cm^2^. All tubes for the liquid flow in the system are made of polypropylene (No. 5392K14, OD = 3/8′, ID = 1/4′, McMaster-Carr, Aurora, OH, USA). The length of tubing on the draw side is 2.0 m and on the feed side 1.8 m. The circulation rates at the draw side and the feed side of the membrane were controlled independently using pumps P1 and P2 along with the flow meters FM1 and FM2, respectively. The draw and feed sides were maintained at the same pressures (atmospheric pressure), which was verified by the pressure gauges PG1 and PG2, respectively. The system’s temperature was maintained constant using an isotherm refrigerated/heated bath circulator (Model 4100 R20, Fisher Scientific, Waltham, MA, USA) and the heat exchangers HEX1 and HEX2. The water transfer across the membrane was monitored using the balances B1 and B2, while the conductivity/temperature meter T-C monitored the salt transfer. The B1, B2, and T-C were connected to a personal computer, and the experimental data was recorded using a LabView data acquisition program. The details of equipment shown in Figure 2 are summarized in Table 1.

After installing in the membrane cell, each membrane coupon was thoroughly cleaned by circulating DI water at the feed and draw sides of the membranes. Once the conductivity of the circulating water reached a constant value, the feed and draw solution tanks were charged with fresh DI water. The cleaning continued until the conductivity at both sides of the membrane reached the minimum detectable value of 30 µs/cm, requiring several fresh DI water batches. Then, the system was stopped, and after replacing the DI water in the draw tank with the draw solution of the desired concentration, it restarted in a bypass mode circulating the draw and feed solutions at 2.4 L/min until and running until the set temperature of 25 °C was reached. The experiment was initiated by re-directing the draw and feed solutions from the respective bypass lines (BP1, BP2)to the draw and feed sides of the membrane cell. Re-directing the streams took less than 2 s. The mass and concentration of the feed and draw solutions were recorded every 2.5 s before and during the experiment. The former ensured stabilization of the system (constant temperature and constant mass of the draw and feed solutions) before initiating the experiment. Each experiment was carried out for 30 min to ensure reaching pseudo-steady-state. The processing of the experimental data included correcting for the disturbance caused by re-directing the streams and possible evaporation effects [20].

The experiments were carried out with the AL-DS and AL-FS orientations using the following three different concentrations of the draw solution: 1 mol/L, 2 mol/L, and 4 mol/L, which were prepared by dissolving sodium chloride (Fisher Scientific, Waltham, MA, USA, purity > 99.5%) in DI water. The latter was produced by a Super-Q Plus Water Purification System (Millipore Sigma, Burlington, MA, USA). The feed solution in all experiments was DI water. When the draw tank was charged with a draw solution, all tubes in the system and the membrane cell were filled with DI water. This led to some dilution of the draw solution so that the actual experimental concentrations of the draw solution were less than the values listed above. Three experiments with three different coupons cut from the same commercial membrane were performed at each condition. Therefore, 18 different coupons were tested at 6 different conditions. 

## 4. Results and Discussion

### 4.1. Effect of Membrane Orientation

Figure 3 presents the progress of a dynamic experiment with the AL-DS orientation of the membrane. Initially, in contact with DI water, the active layer was suddenly exposed to a 2 M aqueous solution of NaCl. Figure 3a presents the normalized mass of the draw and feed solutions as a function of time and the corresponding total mass. The normalized mass equal to zero corresponds to the initial mass of the respective solutions, which for the feed and draw were 400 g and 700 g, respectively. Figure 3b presents the corresponding change in the mass of NaCl in the feed solution, which was determined from the product of the mass of the feed solution and the concentration of NaCl based on the measured conductivity. The water flux (*J_w_*) and salt flux (*J_s_*) are directly proportional to the respective mass change rates in Figure 3.

It is evident from Figure 3a that the highest water flux rate occurs immediately after the initiation of the experiment, and it gradually decreases until it becomes constant, after approximately 5–6 min. On the other hand, the rate of the salt flux is zero in the first couple of minutes, after which it gradually increases until reaching a constant value 7–8 min from the beginning of the experiment. The constant mass increase/decrease rate in Figure 3 indicates attaining pseudo steady-state conditions, which we refer to as steady-state conditions. The straight lines in Figure 3 were obtained by linear regressions of the data collected between 10 and 30 min, i.e., when the system was clearly at a steady state. The steady-state *J_w_*s based on the linear regression line slope at the feed side and draw are 6.48 L/m^2^·h and 6.34 L/m^2^·h, respectively. A difference between the two steady-state water fluxes, which is likely due to the evaporation effects, indicates the system’s limitations [20]. When reporting the steady-state water flux, we will use the arithmetic average of the two values. Therefore, for the membrane depicted in Figure 3a, *J_w_* = 6.41 L/m^2^·h.

In addition to the steady-state fluxes, the linear regression lines in Figure 3 determine the time lags associated with water and salt transport across the membrane, respectively. The time lag is the intersection of the extrapolated straight line corresponding to steady-state flux with the time axis. Consequently, the time lags for the water transport (*θ_w_*) based on the draw and feed solutions are −11.5 min and −12.3 min, respectively. In principle, these time lags should be the same. The observed difference arises from the previously mentioned slight difference between the steady-state water fluxes and uncertainty in the normalized mass after the step change [20]. The average water transport time lag in Figure 3a is, therefore, *θ_w_* = −11.9 min. The shape of the mass of water vs. time curve in The shape of the pressure response in Figure 3a is similar to a pressure decay curve due to gas transport into a membrane in a dynamic gas permeation (time-lag) experiment [21]. 

The steady-state salt flux and the corresponding time lag for the membrane in Figure 3b are *J_s_* = 12.50 g/m^2^·h and *θ**_s_* = 4.16 min. The mass of NaCl vs. time curve resembles a pressure rise curve in a dynamic gas permeation (time-lag) experiment [28]. Considering the time lags of water and salt in Figure 3, *θ_w_* = −2.8*θ_s_*. On the other hand, based on Equations (14) and (15), *θ_u_* = −2*θ_d_*. It is important to emphasize that although Figure 3a resembles a pressure decay curve, Figure 3b pressure rise curve does not mean that *θ_w_* should be −2*θ_s_*. Unlike a model time-lag experiment involving a homogeneous gas permeation membrane for which Equations (14) and (15) are applicable, a dynamic FO experiment involves at least two resistances in series and two different species, solvent and solute, which move in opposite directions.

Figure 4 presents the progress of a dynamic experiment with the AL-FS orientation of the membrane. Initially, in contact with the DI water, the membrane’s porous sublayer was suddenly exposed to a 2 M aqueous solution of NaCl. The only difference between the experiments depicted in Figure 3 and Figure 4 is the membrane orientation.

The behavior in the experiment with the AL-FS orientation (Figure 4) appears to be very different from the experiment in the AL-DS orientation (Figure 3). There is practically no dynamics in water transfer across the membrane; the rate of mass loss at the feed side and the mass gain at the draw side appear constant right from the beginning of the experiment. The *J_w_* based on the mass loss at the feed side and the mass gain at the draw side, determined using the time frame from 10 min to 30 min, are 3.85 L/m^2^·h and −3.81 L/m^2^·h, respectively, or the average *J_w_* = 3.83 L/m^2^·h. It is roughly half of *J_w_* in the experiment with the AL-DS orientation. Extrapolating the best-fitted straight lines between 10 min and 30 min to the time axis leads to the respective water time lags of close to −0.7 min. Because of the relatively low *J_w_* in the AL-FS orientation, the data in Figure 4a is more scattered than in Figure 3a due to the limited resolution of the balances. In turn, this scatter affects the experimentally determined *θ_w_*. Consequently, small water time lags (−1 min <*θ_w_* < 1 min) are associated with considerable uncertainty. In other words, the current resolution of the system does not allow accurate observation of the dynamics in experiments such as the one shown in Figure 4a.

Interestingly, there appears to be a dynamic behaviour associated with the salt transport in Figure 4b. Based on the slope of the mass gain of NaCl in the period from 10 min to 30 min, *J_s_* = 3.23 g/m^2^ h and *θ_s_* = 4.8 min, respectively. Therefore, *J_s_* in the AL-FS orientation is almost four times lower than in the AL-DS orientation. On the other hand, *θ_s_* in both membrane orientations are comparable. 

The ratio of the water and the reverse salt fluxes is a measure of membrane selectivity; more specifically, it is referred to as the reverse solute flux selectivity (RSFS) [29]. The RSFS is 0.51 L/g and 1.19 L/g in the AL-DS and AL-FS orientations. Therefore, the membrane that operated in the AL-FS orientation is more selective than in the AL-DS orientation. The RSFS in both membrane orientations should be the same [10]. However, this is not the case for many membranes reported in the literature. For example, Ghanbari et al. [30] observed a similar relation between the RSFS in the AL-DS and AL-FS orientations. On the other hand, Wang et al. [31] and Li et al. [32] reported the RSFS in the AL-DS orientation to be greater than that in the AL-FS orientation.

Although *θ_s_* in both orientations are comparable (4.2 min vs. 4.8 min), the progress of the respective dynamic experiments is different. In Figure 4b, NaCl appears in the feed solution right after the experiment’s initiation. Moreover, the salt transfer rate in the first 6–7 min seems to be constant, after which it quickly transforms to another constant rate, which is greater than the first one. Therefore, the shape of the curve in Figure 4b does not resemble the pressure rise curve in a dynamic gas permeation (time-lag) experiment [28].

### 4.2. Physical Interpretation of Dynamic Experiments

The progress of the dynamic experiments shown in Figure 3 and Figure 4 can be explained by using the simplified structure of a TFC membrane presented in Figure 5. The blue and red arrows indicate water and salt transport. The membrane consists of an active layer L1 and a porous sublayer L2. The active layer is imperfect, i.e., it contains nonselective pores, which are considered membrane defects. It is important to emphasize that commercial RO membranes inevitably contain defects induced during the synthesis process. These defects allow for solution leakage during the membrane operation [33]. In other words, the transport in the membrane depicted in Figure 4 would follow the SDI model described by Equations (4) and (5). It is also essential to remember that the membrane must be selective for a concentration gradient to generate an osmotic pressure gradient [8].

Following the step change in the AL-DS orientation (Figure 3), the concentrated draw solution instantly contacts the active layer. In contrast, the other side of the active layer is exposed to DI water that fills the pores of the support layer. Therefore, immediately after the initiation of the experiment, there is the highest osmotic pressure gradient across the active layer and hence the maximum water flux. At the same time, the salt starts to diffuse through the nonporous part and the defects in the active layer. Once the salt appears at the other side of the active layer, the osmotic pressure gradient decreases, decreasing the water flux, which is evident in Figure 3a. If there were no defects in the active layer, it would take some time before the solute molecules appeared on the other side of the active layer. During that time, the water flux would be constant. However, this is not the case in Figure 3a.

On the other hand, while the water flux continuously decreases in Figure 3a, it takes some time for the solute molecules in the feed solution to appear (Figure 3b). In this period, the salt concentration in the porous sublayer in contact with the active layer increases and the concentration profile in the pores of the sublayer layer is developing. Once a constant concertation profile in the porous sublayer is developed, the rate of solute transport becomes constant, which, according to Figure 3b, occurs approximately 7–8 min from the initiation of the experiment. On the other hand, considering Figure 3a, it takes 5–6 min for the water flux to become constant. It suggests that the salt concentration in contact with the active layer might already be close to the final steady-state value, whereas the concentration profile downstream from the selective layer would continue to develop.

The experiment in the AL-FS orientation confirms the presence of the defects in the active layer. As shown in Figure 4, the water flux appears to be constant right after the initiation of the experiment. It suggests that the concentrated draw solution contacts the active layer almost instantaneously after the step change. A constant salt flux right after the initiation of the experiment further confirms immediate contact of the draw solution with the active layer via the porous sublayer. Should there be only one diffusion path for salt, the salt flux would be constant throughout the entire experiment. However, this is not the case in Figure 4b. Between 7–8 min from the initiation of the experiment, the slope in Figure 4b increases to a new higher value. It is because of the appearance of solute molecules diffusing through the non-defective part of the active layer on the other side of the membrane. In other words, the first constant slope in Figure 4b is likely due to the leakage of the draw solution through the active layer of the membrane. The difference between the final slope and the initial slope in Figure 4b would correspond to the rate of salt diffusion through a non-defective part of the active layer.

### 4.3. The Effect of Draw Solution Concentration

As previously stated, three different concentrations of the draw solution were investigated, namely, 1 M, 2 M, and 4 M. For each draw solution concentration, six independent tests, three in the AL-DS orientation and three in the AL-FS orientation, were performed. Therefore, 18 dynamic experiments were carried out using 18 different membrane coupons obtained from the same commercial membrane. A summary of the experimental results is presented in Table 2. For the water transport, *J_w_* and *θ_w_* listed in Table 2 are the averages of the respective values based on the water gain at the draw side and the water loss at the feed side. It can be noticed that there is a considerable variation, in particular in the water and salt time lags, from coupon to coupon in Table 2. Significant variability in FO performance between different coupons from the same membrane is commonly observed even for commercial membranes [34]. 

As the draw solution concentration increases, *J_w_* and *J_s_* increase for both membrane orientations. It was expected because of the increase in the driving force for water and salt transport. However, the relative increase is considerably different for water and salt; it differs depending on the membrane orientation. For example, as the draw solution concentration increases from 1 M to 4 M, *J_w_* in the AL-DS and AL-FS orientations increases by 13% and 41%, respectively. The corresponding *J_s_* in the two orientations increases by 216% and 273%. The *J_w_* is higher in the AL-DS orientation than in the AL-FS orientation at any draw solution concentration. Still, *J_s_* in the former orientation is much greater than in the latter. Considering the RSFS, it is clear that at any concertation of the draw solution, the operation in the AL-FS orientation results in a greater RSFS, which is consistent with the previously discussed observation based on Figure 3 and Figure 4. Moreover, the RSFS in both orientations decreases as the draw solution concentration increases. 

The effect of the draw solution concertation on the water and salt lags depends on the membrane orientation. Increasing the draw solution concentration from 1 M to 4 M in the AL-DS orientation doubles *θ_w_* and *θ_s_*. On the other hand, the draw solution concertation has little effect on the respective time lags in the AL-FS orientation. Moreover, the water time lag in this orientation is a slightly negative value, which becomes closer to zero when the concertation of the draw solution increases. The salt time lag in the AL-FS orientation ranges between 4.39 and 4.63 min, which is a slight variation. It is important to emphasize that the shape of the mass of NaCl vs. time curve in all FO experiments was similar to that shown in Figure 4b. 

To explain the effect of membrane orientation and the draw solution concentration on the water and salt time lags, we recall that in the AL-FS orientation, the water flux is practically constant during the entire experiment (Figure 4a). At the same time, reverse salt flux is also constant, but after several minutes, it transforms to another constant value (Figure 4b). It further confirms that the draw solution must quickly contact the selective layer. Moreover, a near steady-state salt concentration in contact with the selective layer must be established shortly after the initiation of the experiment. Consequently, the resistance to salt diffusion in the active layer primarily determines the salt time lag in the AL-FS orientation. In other words, the resistance to salt diffusion in the porous sublayer in the AL-FS orientation (internal concertation polarization) does not influence the observed salt time lag.

On the other hand, in the AL-DS orientation, the time lag of both water and salt increases as the draw solution concentration increases. Moreover, the salt time lag in the AL-DS orientation is generally higher than in the AL-FS orientation. As the draw solution concertation increases, the water flux increases. Salt species must diffuse the water-filled pores of the sublayer when water moves in the opposite direction. This is probably why the time required to reach a steady-state concentration profile in the porous sublayer and hence the salt time lag increase with the concentration of the draw solution. In other words, unlike in the AL-FS orientation, the internal concertation polarization affects the salt time lag in the AL-DS orientation.

This study’s new characterization method of TFC membranes provides much new information about the studied membrane. Assuming that the resistance to salt diffusion in the defect-free part of the active layer is independent of the salt concentration, which appears to be a reasonable assumption, the difference between the salt time lags in the AL-DS and AL-FS orientations could be a measure of the internal concertation polarization. In turn, the structure parameters of the membrane influence the internal concertation polarization. Suppose the internal concertation polarization does not affect the salt time lag in the AL-FS orientation. In that case, this time lag is directly proportional to the thickness of the selective layer and inversely proportional to the diffusivity of the solute in the selective layer. Moreover, the difference between the steady-state reverse salt flux and the early constant reverse salt flux in the AL-FS orientation could allow decoupling the salt transfer through the defects in the selective layer from the transport in a nonporous part of the selective layer.

## 5. Conclusions

Dynamic forward osmosis tests initiated by a step-change in the draw solution concertation is a new approach to characterize osmotic membranes. This work systematically studied the effects of membrane orientation (AL-DS vs. AL-FS) and the magnitude of change in the draw solution concentration on the dynamic and steady-state behaviour of commercial low-pressure RO TFC membranes. For a given concentration of the draw solution, the membranes tested in the AL-DS orientation have a greater water flux and a significantly greater reverse salt flux than the same membranes tested in the AL-FS orientation. Consequently, the reverse solute flux selectivity in the AL-FS orientation is higher than in the AL-DS orientation. Moreover, while the water flux and the reverse salt flux increase with the draw solution concentration in both membrane orientations, the increase in the reverse salt flux is more significant than that in the water flux. As a result, both modes’ reverse solute flux selectivity decreases with the draw solution concentration increase.

The analysis of the results from the dynamic tests revealed significantly different behaviors of the same membrane when tested in the AL-DS and AL-FS orientations. There is an apparent negative time lag associated with the water flux in the AL-DS orientation, ranging from −10 to −20 min. The absolute value of the water time lag increases with the increase in the draw solution concertation. On the other hand, the water flux in the AL-FS mode appears to reach a steady-state value right after the initiation of the experiment, and the resulting time lag is close to zero regardless of the draw solution concertation. Although we observed the salt time lag in both modes, the behavior in the AL-DS orientation is markedly different from that in the AL-FS orientation. More specifically, in the AL-DS orientation, the salt does not appear in the feed solution for at least a couple of minutes, after which the reverse salt flux gradually increases to the steady-state value. On the other hand, in the AL-FS orientation, the salt appears in the feed solution right after the initiation of the tests, and the corresponding reverse salt flux is constant. However, after several minutes, the reverse salt flux quickly transforms to a new, higher steady-state value. 

Considering transient and steady-state behaviour in the AL-DS and AL-FS orientations at different draw solution concentrations, the new characterization method allows qualitatively assessing several important membrane characteristics related to the active and porous sublayer of the membrane. In particular, the difference between the salt time lags in the two orientations at a given draw solution concertation could measure the structural parameter. Two different reverse salt flux rates in the AL-FS orientation could be used to assess the quality of the active layer of the membrane. Finally, the salt time lag in the AL-FS orientation could be used to measure the resistance to salt diffusion in the selective layer of the membrane. The main challenge of the dynamic method is the limited resolution of the balances that measure the rate of mass change. As a result, short time lags (one minute or less) are associated with considerable uncertainty. However, even with this limitation, the dynamic forward osmosis tests offer a potent membrane characterization tool, which could be very useful for developing new membrane materials and new synthesis protocols for improved osmotic membranes.

## Figures and Tables

**Figure 1 membranes-12-00385-f001:**
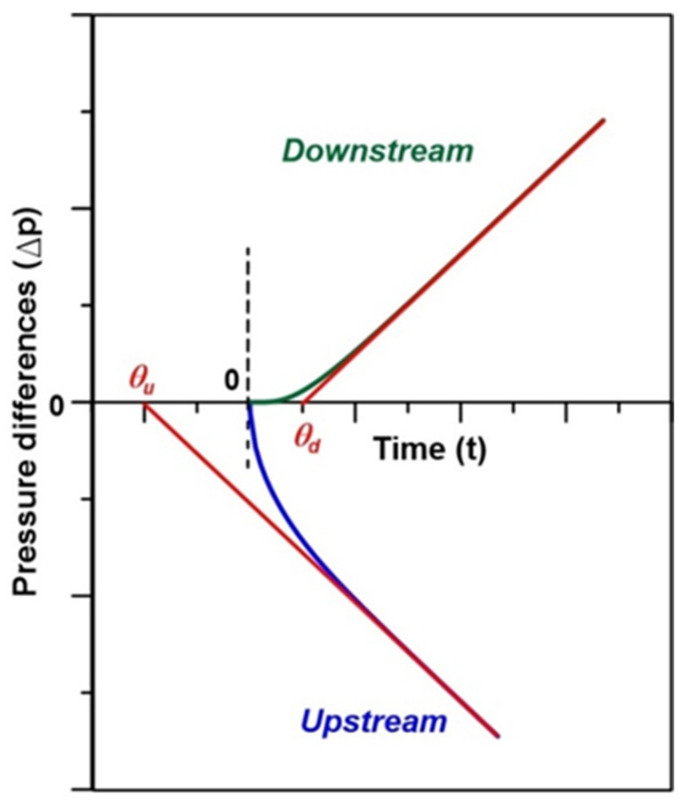
Progress of dynamic gas permeation experiment based on the pressure rise downstream (top curve) and pressure decay upstream (bottom figure) [26]. Reprinted with permission from Ref. [20]. 2020, Elsevier.

**Figure 2 membranes-12-00385-f002:**
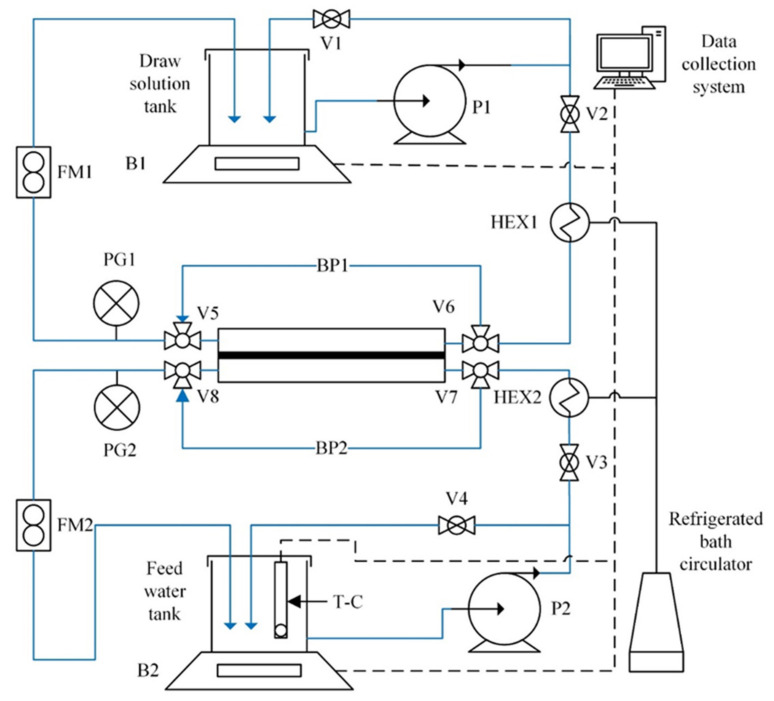
Schematic diagram of the FO testing setup. Adapted with permission from Ref. [20]. 2020, Elsevier.

**Figure 3 membranes-12-00385-f003:**
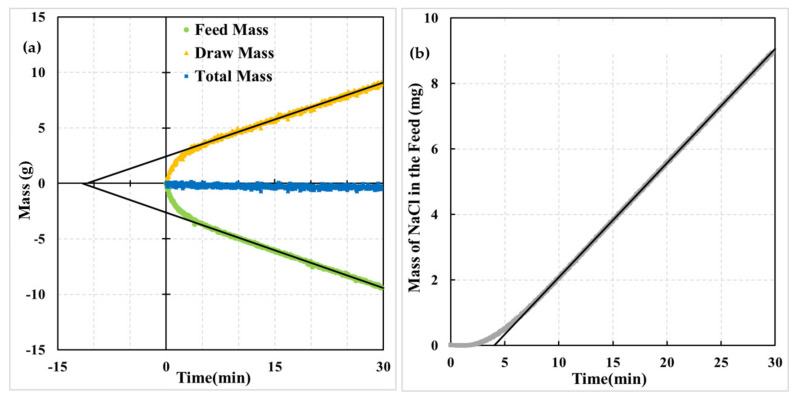
Progress of an experiment with the AL-DS membrane orientation; step change in the draw so-lution concentration of NaCl from 0 to 2 M; experiment carried out at 25 ± 0.3 °C; (**a**) mass of the feed, draw, and feed + draw as a function of time; (**b**) mass on NaCl in feed solution as a function of time. Adapted with permission from Ref. [20]. 2020, Elsevier.

**Figure 4 membranes-12-00385-f004:**
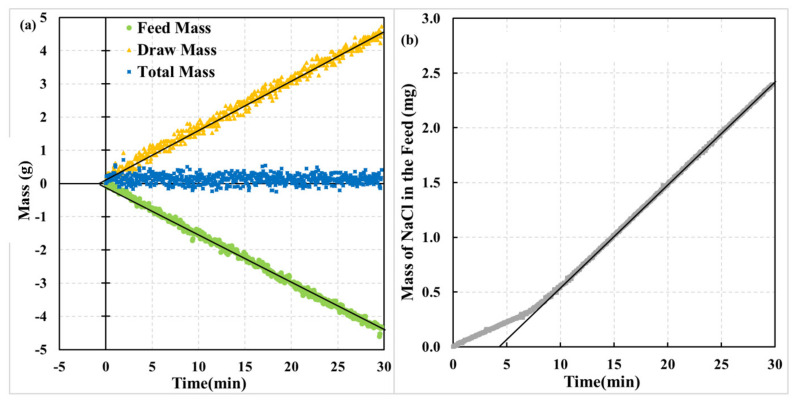
Summary of an FO experiment; step change in the draw solution concentration of NaCl from 0 to 2 M/L; experiment carried out at 25 ± 0.3 °C; (**a**) mass of the feed, draw, and feed + draw as a function of time; (**b**) mass on NaCl in feed solution as a function of time.

**Figure 5 membranes-12-00385-f005:**
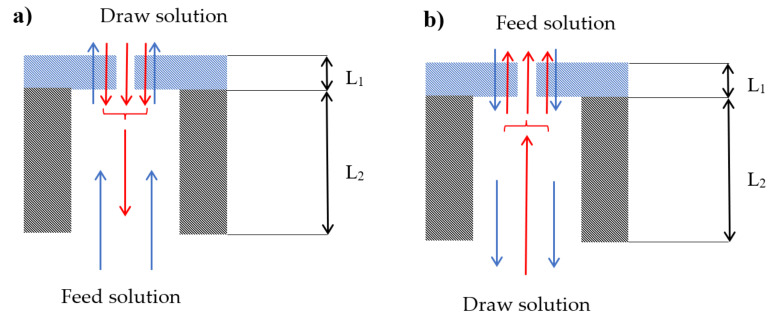
Schematic representation of TFC membrane and the transport of water and salt in (**a**) AL-DS orientation and (**b**) AL-FS orientation.

**Table 1 membranes-12-00385-t001:** Specifications of the components in the FO testing system.

Componet	Description	Supplier
V1–V4	On/off valves, 4757K18	McMaster-Carr, USA
V5–V8	Diverting ball valves, 4757K62	McMaster-Carr, USA
FM1, FM2	0–4 LPM flowmeters	Blue-White Indust., USA
PG1, PG2	0–3 psig pressure gauges	McMaster-Carr, USA
B1, B2	0.01 g resolution balances, 6202-1S	Entris Precision, Germany
P1, P2	Centrifugal pumps P1 and P2, TE-3-MD-HC	Little Giant Co., USA
T-C	Conductivity/temperature meter, CON2700	Oakton Instruments, USA

**Table 2 membranes-12-00385-t002:** Summary of dynamic experiments with Toray low-pressure membrane at 25 ± 0.3 °C. The effect of the draw solution concentration in AL-DS and AL-FS membrane orientations.

Concentration Draw Solution (M)	Orientation	Membrane	Water Transport		Salt Transport	
			Flux ^1^ (L/m^2^·h)	Time Lag ^1^ (min)	Flux (g/m^2^·h)	Time Lag (min)
1	AL-DS	M1	5.11	−9.58	7.08	3.50
		M2	6.65	−9.27	6.79	3.58
		M3	4.02	−14.61	8.27	4.30
		**Average**	**5.26**	**−11.15**	**7.38**	**3.79**
1	AL-FS	M4	4.08	−0.15	2.39	4.83
		M5	3.57	−0.35	2.75	2.93
		M6	4.19	−1.72	2.03	5.23
		**Average**	**3.98**	**−0.74**	**2.38**	**4.33**
2	AL-DS	M7	6.41	−11.92	12.50	4.16
		M8	5.40	−26.12	7.54	7.58
		M9	4.80	−18.65	12.55	4.97
		**Average**	**5.43**	**−18.90**	**10.86**	**5.56**
2	AL-FS	M10	3.83	−0.69	3.23	4.82
		M11	4.61	0.07	3.07	4.91
		M12	4.27	−0.79	3.24	4.18
		**Average**	**4.24**	**−0.47**	**3.18**	**4.63**
4	AL-DS	M13	7.35	−25.81	10.50	8.26
		M14	6.06	−15.17	27.84	3.92
		M15	5.86	−19.09	22.10	5.37
		**Average**	**6.42**	**−20.03**	**20.12**	**5.85**
4	AL-FS	M16	5.61	−0.57	1.33 ^2^	0.79 ^2^
		M17	5.41	−0.18	5.96	4.77
		M18	5.83	0.47	4.21	4.00
		**Average**	**5.62**	**−0.09**	**5.09**	**4.39**

^1^ Average from the respective values based on water loss in the feed solution and water gain in the draw solution. ^2^ Not taken into consideration for calculating the average values.

## Data Availability

Not applicable.

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
