# Peer review of "Effect of Membrane Orientation and Concentration of Draw Solution on the Behavior of Commercial Osmotic Membrane in a Novel Dynamic Forward Osmosis Tests"

_membranes, 2022, doi:10.3390/membranes12040385_

Round 1

Reviewer 1 Report

Review: membranes-1652378

Title: Effect of membrane orientation and concertation of draw solution on the behavior of commercial osmotic membrane in a novel dynamic forward osmosis tests

The article analyzes the effect of membrane orientation and concertation of draw solution on the behavior of commercial osmotic membrane in novel dynamic forward osmosis tests. I understand that this study is interesting and has adherence with the research published by the Membranes MDPI. Nevertheless, some adjustments must be made before this occurs. The most important points from this review are described below.

Title: Maybe there was a spelling problem in the title of the document. Did you actually mean to refer to concertation (instead of concentration) of draw solution?

Abstract: As one would expect from this session, the Abstract presents, in a summarized way, all the elements that make up the study. The problem, as well as the results and conclusions, were well described. However, there is no information (or at least an indication) of how the analyzes were carried out. I suggest that this content be presented in 1 or 2 lines.

General comments: It would be very convenient for each information cited in the manuscript, which had been collected from the literature, to receive its corresponding citation. The compression of citations (as in line 30) or their indication in an integrated manner (Lines 35, 41) makes the text difficult to understand. So, I suggest reviewing this form of presentation.

Introduction: Lines 36-53: This content seems more suitable for the following section of the document (Theoretical Backgrounds) than for the Introduction to the study. Perhaps this passage could be condensed to keep the text flowing.

Another aspect that stands out in the Introduction is the degree of detail of each study cited to define the research gap. By proceeding in such a deep and rhetorical way, the article loses agility, and with that, there is a risk of the reader getting dispersed. I suggest that this content (which comprises Lines 54-116) be revised to become more objective. Some of the data and information communicated there could be transferred to the Results and Discussion session, even improving this element. This remodeling would make the Introduction shorter and more objective, opening space for the eventual placement of other references on the topic and making it more concise.

Theoretical Backgrounds: The content of this session was written in an appropriate and even didactic way. In this case, the care to be observed is to ensure that all the elements indicated in the Theoretical Backgrounds are being used in other sections of the manuscript. As far as I can see, this connection is not perfectly established. Therefore, I suggest that a brief review of the content be done.

Materials and Methods: Line 237: Conceptually justify the choice of Low-pressure RO TFC membranes from Toray Industries to carry out this study.

Lines 240-241: The (detailed) scheme of the testing system is an important element of the working method. The same is true of the experimental protocol of dynamic experiments (Line 253). When this does not occur, even the review of the manuscript's content may be impaired. Therefore, this element must be reported with adequate rigor and detail in this section of the document.

On the other hand, I felt a lack of more information about the verified parameters and, perhaps mainly, how these surveys corroborate the objective of the research (this link is fundamental!). There is also no reference to how you dealt with methodological and experimental uncertainties and how these propagate and influence the results obtained. These explanations and additions should be added.

Lines 279-284: This passage more closely resembles the description of the Working Method than the actual results obtained. Although associated with Figure 3, I am not sure that this content helps to clarify the results indicated in this schema. Therefore, this content also needs to be reviewed and/or replaced.

Lines 297-299: In the excerpt: '(...) The straight lines in Fig. 3 represent the respective linear regressions of the data collected in the period between 10 and 30 min, that is when the system was clearly in a steady state. (...)', wouldn't this behavior be expected? Why?

Line 314-315: To approach the phenomenon by Henry's Law suggests that there is an ideal thermodynamic behavior of the chemical species involved in the system. Would this not be a particularization that limits the model's applicability?

Line 315-316: Same as the previous comment to apply the assumption that the gaseous diffusion coefficient is constant.

Lines 317-319: I encourage authors to discuss further the fact that the absolute mean time interval of water to the time interval of salt is 2.8 (therefore > 2.0). Should the system behave this way? What are the implications of this deviation, mainly in the efficiency of the arrangement?

Line 440-441: Wouldn't it already be expected that The effect of the draw solution concentration on the water and salt time lags would depend on the membrane orientation? This justification appears in the passage indicated between lines 450-469. What else did this stage of analysis make you realize?

Conclusions: I missed a passage describing the limitations of the study other problems that might affect the results obtained and increase their uncertainties. Please comment on such points.

Reviewer 2 Report

membranes-1652378

This paper evaluates the effect of membrane orientation and concertation of draw solution on the behavior of commercial osmotic membrane in a novel dynamic forward osmosis tests. The topic falls within Membranes MDPI scope and addresses relevant issues in the osmotically driven membrane filtration process. The proposed characterization method is novel and interesting. The research was done systematically, and the manuscript is very well written. A few comments below can be addressed by the authors to enhance the manuscript quality.

  • Some quantitative data are needed in the abstract. It also needs some elaboration to be well understood since the concept of time lag is very new. A reader may find difficulties.
  • The introduction provides a comprehensive review of FO membrane characterization method. However, a work from Fane’s group is missing from the assessment (see: 1016/j.jwpe.2018.06.011)
  • Please provide briefly the experimental protocol of dynamic experiments. The paper must be self-sufficient and does not largely depend on another paper to be understood.
  • How batch-to-batch variations would affect the analysis? Data in Table 1 show significant variation between membrane samples. Meanwhile, highly precise measurements are needed for the water and salt masses to fit with the proposed model.
  • Finally, please provide a comparison with the finding using the proposed method and some proposed methods available in the literature on the obtained FO membrane characteristics. There are few reports that also employed the same FO membrane used by the author in this work. What insight on material characteristics can be deduced from the time lag?
  •  
